# Social determinants of tobacco use among tribal communities in India: Evidence from the first wave of Longitudinal Ageing Study in India

Jogesh Murmu[1], Ritik Agrawal[1], Sayantani Manna[1], Sweta Pattnaik[1], Shishirendu Ghosal[1], Abhinav Sinha[2], Ardhendu Sekhar Acharya[1], Srikanta Kanungo[1]*, Sanghamitra Pati[1]*

1 ICMR-Regional Medical Research Centre, Bhubaneswar, Odisha, India, 2 Health Technology Assessment in India (HTAIn), ICMR-Regional Medical Research Centre, Bhubaneswar, Odisha, India

☉ These authors contributed equally to this work.
* drsanghamitra12@gmail.com (SP); srikantak109@gmail.com (SK)

**Data Availability Statement:** The dataset analysed during the current study is available in the LASI

## Abstract

### Background

Evidence on tobacco use among indigenous communities is scarce with available literature based either on a specific region or a particular tribe. Considering the large tribal population in India, it is pertinent to generate evidence on tobacco use among this community. Using nationally representative data, we aimed to estimate the prevalence of tobacco use and assess its determinants and regional variations among older tribal adults in India.

### Methods

We analysed data from Longitudinal Ageing Study in India (LASI), wave-1 conducted in 2017–18. A sample of 11,365 tribal individuals aged ≥ 45 years was included in this study. Descriptive statistics was used to assess the prevalence of smokeless tobacco (SLT), smoking, and any form of tobacco use. Separate multivariable regression models were executed to assess the association of various socio-demographic variables with different forms of tobacco use, reported as adjusted odds ratio (AOR) with 95% confidence interval.

### Results

The overall prevalence of tobacco use was around 46%, with 19% of smokers and nearly 32% smokeless tobacco (SLT) users. Participants from the lowest MPCE quintile group had a significantly higher risk of consuming (SLT) [AOR: 1.41 (95% CI: 1.04–1.92)]. Alcohol was found to be associated with both smoking [AOR: 2.09 (95% CI: 1.69–2.58)] and (SLT) [AOR: 3.05 (95% CI: 2.54–3.66)]. Relatively higher odds of consuming (SLT) were associated with the eastern region [AOR: 6.21 (95% CI: 3.91–9.88)].

data repository held at ICT, IIPS [https://g2aging.org/?section=overviews&study=lasi].

**Funding:** The author(s) received no specific funding for this work.

**Competing interests:** The authors have declared that no competing interests exist.

## Conclusion

This study highlights the high burden of tobacco use and its social determinants among the tribal population in India, which can help tailor anti-tobacco messages for this vulnerable population to make tobacco control programs more effective.

## Introduction

Globally, tobacco use is one of the greatest public health threats. The rampant use of various tobacco products is a matter of concern in low-and middle-income countries (LMICs). Evidence suggests that almost 1.3 billion people use tobacco globally, out of which 80% live in LMICs, where tobacco-related morbidities and mortality are highest [1]. Among LMICs, India is the second-largest consumer and still a large-scale producer of tobacco products [2,3]. The tobacco ecosystem in India is complex [4]. According to Global Adult Tobacco Survey-2 (GATS-2, 2016–17), there were nearly 267 million tobacco users aged ≥15 years in India, and among them, about 42.4% were men, and 14.2% were women [5].

Tobacco is commonly used in two ways, i.e., smokeless and smoking form. Due to socio-cultural acceptability, smokeless tobacco (SLT) use is highly prevalent in India, which includes chewing tobacco such as *khaini*, *gutkha*, betel quid with tobacco, *mishri*, *gul*, and *gudakhu* [6]. Other forms include smoking (cigarette, *hookah*, and *bidi*) and the use of any form of tobacco, which simultaneously predisposes a risk of oral submucous fibrosis (OSMF), a premalignant disorder with potency to transform into oral cancer [7]. Evidence supports that smoking is responsible for health ailments of the cardiovascular and respiratory systems [6]. Tobacco not only results in loss of lives but also levies associated social and economic costs. In India, the total financial cost of tobacco uses for all illnesses during 2017–18 among people aged ≥35 years is approximately INR 177 billion (US $ 27.5 billion) [8].

India is a diversified land of many cultures and ethnic groups with one of the largest tribal populations in the world. According to the Census of India-2011, the scheduled tribes (STs) comprised around 104 million people, 8.6% of the national population [9]. There are 550 tribes in India, including particularly vulnerable tribal groups (PVTGs). Approximately 90% of the country's tribal population lives in rural areas, while the remaining 10% resides in urban areas [10]. Still, a gap exists between the indigenous and non-tribal people as the former prefer to live in their geographical habitats, which are often secluded in forests and hard-to-reach areas. Their culture and traditions broadly vary, which disintegrates them from mainstream socio-economic activities; a life often marred with subsistence-based existence, invariably leading to their lower education (41% with no formal education) and socio-economic attainment (41% below the poverty line) [11]. This social exclusion often compels these marginalized communities to have disparities in accessing public health services.

Moreover, deleterious habits such as tobacco use are deeply rooted in their social beliefs and cultural practices. It often remains entrenched in their practices which is difficult to mould. As per the social affirmation and traditional practices, the pervasiveness of tobacco consumption is quite apparent. For instance, SLT is wrongly perceived as safer than smoking, resulting in higher consumption, early initiation, and persistence as a norm [12,13]. Among the indigenous communities, mortality, morbidity, and malnutrition rates are still higher than the average Indian population due to various barriers such as language, education, and infrastructural advancements [14].

Since ancient times, the habitual consumption of tobacco has been an eminent practice among tribal communities [15]. Tobacco uses among the general population is widely studied

based on the Global Youth Tobacco Survey (GYTS) and Global Adult Tobacco Survey (GATS). However, it is scarce in the context of the tribal population [16,17]. Most of the evidence on tobacco use among tribes is either from a specific region or based on a survey from a particular tribe with no nationally representative study. Even within tribals, the available evidence largely focuses on adolescents or young adults, with limited literature on the aging population. Considering the large tribal population in India and their unique health-related behaviours, it is pertinent to generate evidence on tobacco use among this community to pave the way for future tobacco control programs and policies targeting this marginalized group. Hence, we aimed to estimate the prevalence of tobacco use, and assess its determinants and regional variations among older tribal adults in India using nationally representative data.

## Materials and methods

### Overview of data

This study is based on the Longitudinal Ageing Study in India (LASI), wave-1, which was full-scale national survey to scientifically investigate health, economics, social determinants, and consequences of population aging in India; conducted from April 2017 to December 2018 by Harvard TH Chan School of Public Health, the University of Southern California, in partnership with the International Institute for Population Sciences (IIPS), Mumbai. Following our primary objective to estimate the prevalence of tobacco use among older adults we selected LASI data for our analysis. Although the Global Adult Tobacco Survey-2 (GATS-2) also provides information on tobacco consumption, however, LASI primarily focuses on older adults, thus providing a large representative sample of participants aged 45 years or older aligned with our objective. LASI captured detailed socio-economic factors, which is another strength over GATS-2. LASI used three instruments for data collection: household survey schedule, individual survey schedule, and community survey schedule. LASI interviewed a sample of 72,250 individuals aged ≥45 years (and their spouses irrespective of age) who provided written consent before the interviews. A multistage stratified area probability cluster sampling design was employed to reach the final observation [18]. The precise method for LASI, wave-1, has been mentioned on the website of IIPS, Mumbai [19]. We followed the Strengthening the Reporting of Observational Studies in Epidemiology (STROBE) guideline to report this study (**S1 Table**) [20].

### Study participants and sample size

An individual survey schedule was employed among 72,250 participants aged ≥45 years. Participants who referred to their caste as "Scheduled tribes" (STs) were included in this study. Following this, the conclusive sample size of 11,365 tribal individuals aged ≥45 years was achieved as per the objective of this study (Fig 1).

### Outcome variable

The outcome variable for this analysis was tobacco use, i.e., smoking, SLT, and any form of tobacco use among the tribal population in India. The tobacco use status was classified based on the combined responses to the following two questions: "Have you ever smoked or used smokeless tobacco?". Those who responded 'yes' were further enquired: "what type of tobacco product have you used or consumed?" with options such as smoking, SLT, and both smoking and SLT. So, when we created the binary variable "smokers" as an outcome variable, both only smokers and dual users were merged; the sub-groups under smokers were smokers and non-smokers (includes SLT users but don't use smoking tobacco and those who don't use any form of tobacco products). Similarly, when we created another binary variable as " SLT users", the

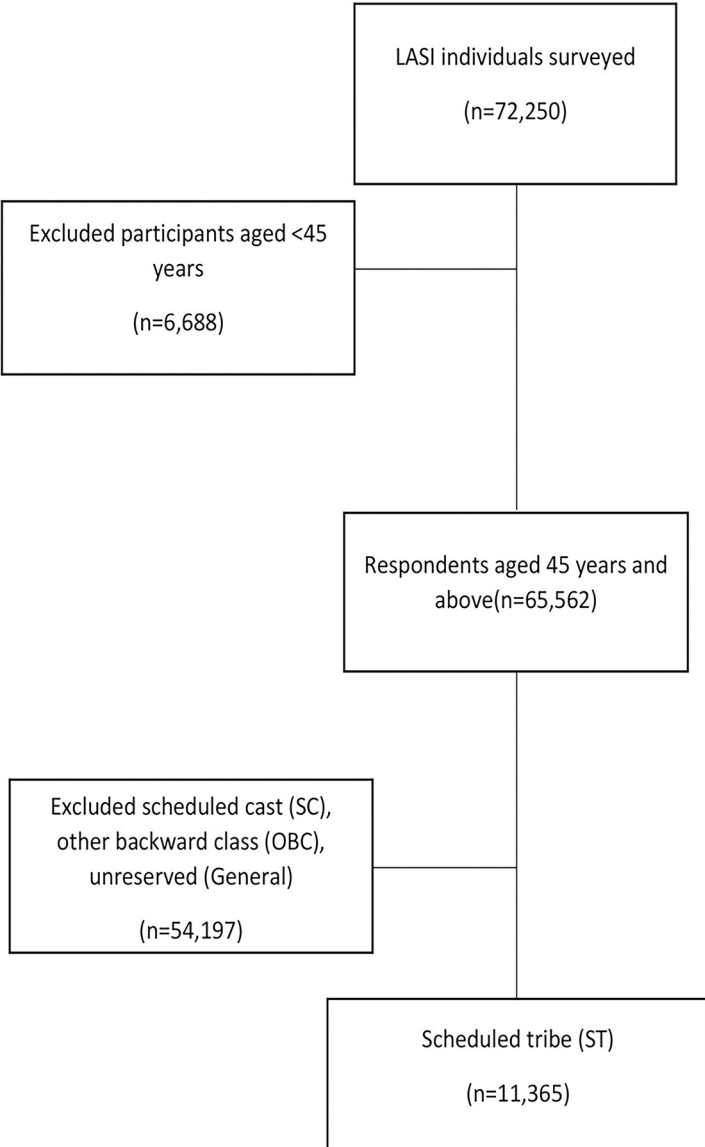

**Fig 1. Selection of study population.**

subgroups under this variable were SLT users (only SLT users + dual users, as they used SLT products also) and non-SLT users (included those who didn't use tobacco of any kind & those who were consumer of smoking tobacco only). Any tobacco group included all forms of tobacco users (only smokers+ only SLT users + dual users) irrespective of the type of products they were using. We conducted three separate logistic regression models for smokers, SLT users and any tobacco users to avoid collinearity.

## Covariates

We employed various socio-demographic variables such as age, gender (male/female), residence (rural/urban), education, occupation, partner status, regions of the country, and wealth index. Additionally, we included two attributes of personal behaviour, i.e., physical activity and alcohol consumption, in the analysis. Age, a continuous variable provided in the LASI

dataset, was divided into three categories according to the LASI report, i.e., 45 to 59 years, 60 to 74 years, and ≥75 years. Previous literature has also categorized age groups in this manner [21]. The education was classified based on the responses to two questions "Have you ever attended school?" Those who answered "no" were categorized as having no formal education, whereas those who answered "yes" were then asked about their "Highest level of education". Respondents who had completed primary school and less than primary were grouped as "up to standard VII". Respondents who completed middle school and secondary school were clubbed as "standard VIII-X". Additionally, individuals who completed secondary school, a diploma, a graduate degree, a postgraduate degree, or a professional degree were classified as "upper secondary and above". Respondents who did not work for more than three months in their lifetime and were not employed were grouped as "currently not working" while the rest were "currently working". We classified living situation into three categories: "living alone", "living with spouse" (which includes people who live with their spouse, others and with children), and "living without spouse" (which includes people who live with children and others; or other people only). MPCE quintiles (lowest, lower, middle, higher, highest) were based on the monthly per capita expenditure of the general population. 29 States and 6 Union Territories (except Sikkim) of India were arranged into six regions (north, central, east, north-east, west, and south) based on their geographical location. Participants who were hardly or never involved in sports or vigorous activities were grouped as "physically inactive." In contrast, others (daily, once a week, more than once a week, as frequent as one to three times a month) were merged into the "physically active" group.

**Statistical analysis.** Before analyzing, the data were filtered for all flagged, missing, and no information cases from LASI. Data were analyzed using STATA, v16·0 (STATA Corp., Texas) for Windows. Descriptive analysis was presented for continuous variables, such as age in mean with standard deviation (SD). The frequency and proportions (n, %) of tobacco use for each sub-group were described for categorical variables. The distribution of tobacco consumption in all three forms was tabulated with socio-demographic and behavioral covariates. 95% confidence interval (CI) for all weighted proportions was reported as a measure of uncertainty. We considered a $p$-value of $< 0.05$ to be significant. Univariate logistic regression was applied to determine the crude odds ratio between distinct outcomes and attributes. All socio-demographic characteristics of the study participants were adjusted in separate multivariable logistic regression models to obtain an adjusted association between outcome variables and various attributes, presented as an adjusted odds ratio (AOR) with 95% CI.

Segregating the LASI data at the state level will lead to very few samples of tribal population in smaller states. Therefore, we presented the regional differences in the prevalence of tobacco use, for the four most populated tribal states (Madhya Pradesh, Odisha, Maharashtra, Rajasthan) based on the census, 2011 data. Sampling weights were considered in the analysis to adjust for the multistage sampling design.

## Ethical consideration

Indian Council of Medical Research (ICMR), New Delhi, and IIPS, Mumbai granted ethical approval for the first wave of the LASI survey. However, we used anonymous secondary data, which is available in the public domain. Hence, there is no participant risk. Appropriate permission to use the dataset was received, and data is being acknowledged wherever entailed.

## Inclusivity in global research

Additional information regarding the ethical, cultural, and scientific considerations specific to inclusivity in global research is included in the (S2 Table)".

## Results

For this analysis, 11,365 scheduled tribe participants aged 45 years and above were included. Their average age was 59.3 (±10·7) years. Most of them belonged to the age group of 45–59 years (54.5%). A nearly equal number of male and female interviewees were included, with a little female predilection (53.7%). Most of the tribal population never went to school (54.6%) and lived in rural settings (77.4%). Most of the participants were physically inactive (52.6%) and worked (56.2%) while interviewed (Table 1).

The overall prevalence of tobacco use was 46.1%, while smoking and SLT use was estimated 18.6% and 31.7%, respectively; tobacco in both forms (dual-use) was consumed by 4.2% of the participants. Smoking and SLT use were more prevalent among males (smoking: 34.8% and SLT: 40.1%) (Fig 2).

The prevalence of smoking and SLT use was estimated the highest (23.7% and 38.4%, respectively) among the population aged 75 years and above. Smoking (24.1%) was most prevalent in tribals from southern India, while the eastern tribal population predominantly used

**Table 1. Sociodemographic characteristics of study population (n = 11,365).**

| Socio-demographic Characteristics | | n (%) |
|---|---|---|
| Age (n = 11,365) | 45 to 59 years | 6192 (54.5%) |
| | 60 to 74 years | 4058 (35.7%) |
| | ≥75 years | 1115 (9.8%) |
| Gender (n = 11,365) | Male | 5256 (46.3%) |
| | Female | 6109 (53.7%) |
| Residence (n = 11,365) | Rural | 8801 (77.4%) |
| | Urban | 2564 (22.6%) |
| Education (n = 11,364) | No formal education | 6205 (54.6%) |
| | Up to standard VII | 2942 (25.9%) |
| | Standard VIII-X | 1633 (14.4%) |
| | Higher secondary & above | 584 (5.1%) |
| Occupation (n = 11,360) | Currently working | 6383 (56.2%) |
| | Currently not working | 4977 (43.8%) |
| Living arrangement (n = 11,365) | Living alone | 418 (3.68%) |
| | Living with spouse | 8,170 (71.89%) |
| | Living without spouse | 2,777 (24.43%) |
| Region (n = 11,365) | North | 367 (3.2%) |
| | Central | 1440 (12.7%) |
| | East | 1257 (11.1%) |
| | Northeast | 4951 (43.6%) |
| | West | 1663 (14.6%) |
| | South | 1687 (14.8%) |
| Physical Activity (n = 11,287) | Active | 5346 (47.4%) |
| | Inactive | 5941 (52.6%) |
| Alcohol use (n = 11,288) | Yes | 3267 (28.9%) |
| | No | 8021 (71.1%) |
| MPCE Quintile (n = 11,365) | Lowest | 3274 (28.8%) |
| | Lower | 2438 (21.5%) |
| | Middle | 2066 (18.2%) |
| | Higher | 1776 (15.6%) |
| | Highest | 1811 (15.9%) |

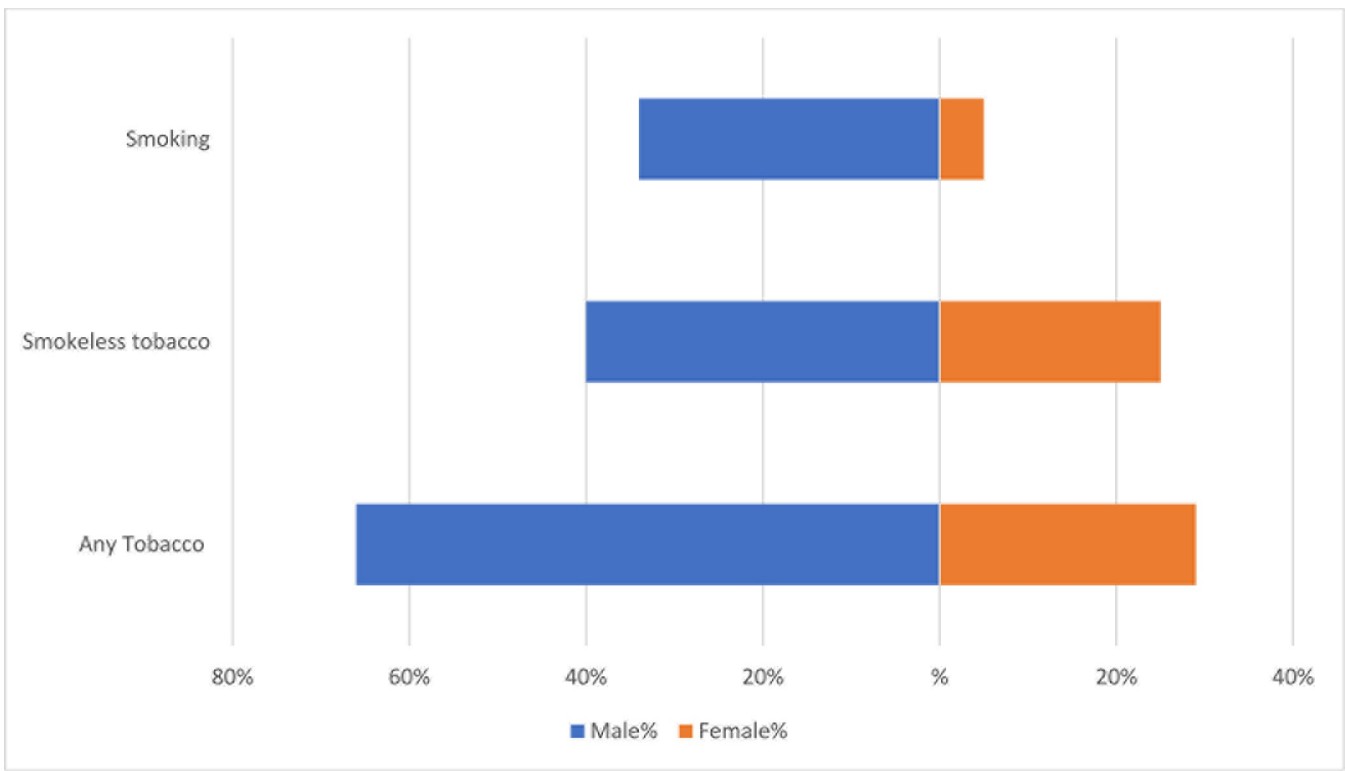

**Fig 2. Type of tobacco consumption across gender among the tribal populations.**

SLT (48.6%). The prevalence of any form of tobacco use among the four most tribal populated states Madhya Pradesh, Maharashtra, Odisha, and Rajasthan were 39%, 43%, 61%, and 46%, respectively (Fig 3). The detailed state-wise prevalence of various forms of tobacco is presented in **S3 Table**.

Tobacco use of any variant was relatively higher among alcohol consumers (smokers: 32.9%, SLT use: 52.5%, and any form of tobacco use: 75.8%). The physically active participants used tobacco of any form more (smokers: 20.0%, SLT use: 36.8%) than those who were not into sports or regular physical work (smokers: 17.3% and SLT use: 26.3%). The detailed distribution of tobacco use is provided in Table 2.

Univariate logistic regression models identified participants aged 75 years and above, males, rural residents with lesser years of schooling, currently working, living with a spouse, and alcohol use were associated with tobacco use (Table 2). Table 3 shows the association of socio-demographic and behavioral factors with different forms of tobacco use among the study population after adjusting for all the covariates. There was a 76% greater likelihood of tobacco use among the 75+ age group (AOR: 1.76 (1.25–2.46). Males were highly associated with using all forms of tobacco (smoking, SLT, or any form) compared to females. Relatively higher odds of smoking were evident from the central [AOR: 1.45 (0.91–2.31)] and southern [AOR: 1.20 (0.73–1.97)] tribal population; however, SLT was highly associated with the eastern [AOR: 6.17 (3.88–9.80)] and western region [AOR: 4.01 (2.49–6.44)]. Alcohol was found to be an attributing factor for any form of tobacco, i.e., smoking [AOR: 2.09 (1.69–2.58)] or SLT [AOR: 3.05 (2.54–3.66)] or any form of tobacco [AOR: 4.06 (3.39–4.87)]. Physical activity was associated with SLT [AOR: 1.33(1.09–1.63)] and any tobacco use [AOR: 1.27 (1.05–1.52)]. The odds of SLT use were 41% higher among the poorest group [AOR: 1.41 (1.04–1.92)].

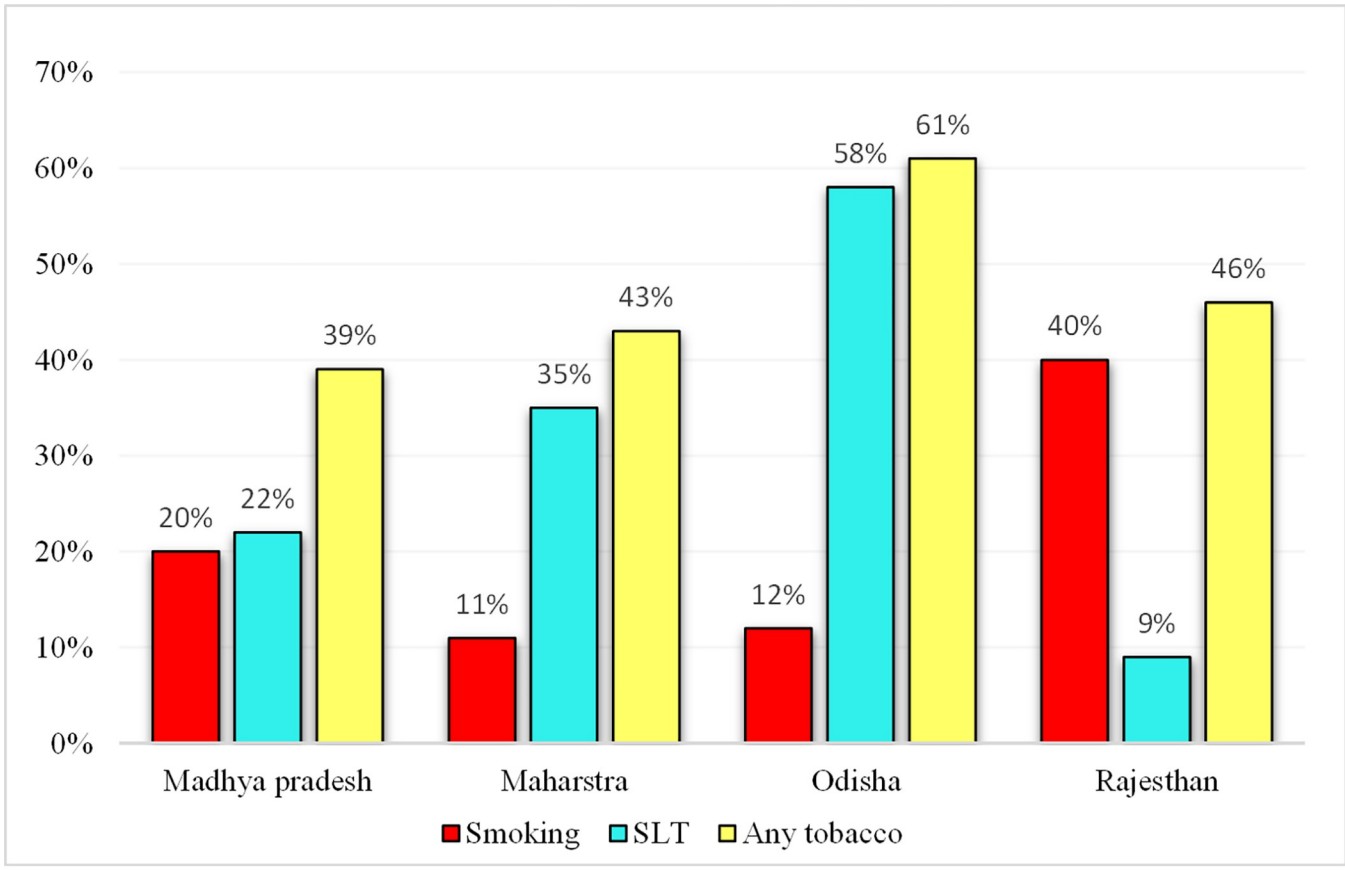

**Fig 3. Prevalence of different forms of tobacco use across major tribal populated states.**

## Discussion

This study reported the prevalence and determinants of tobacco use among older adults from indigenous communities of India using nationally representative data. Overall, 46 percent of tribal population was estimated using any form of tobacco, with 19 percent smokers and approximately 32 percent SLT users. Those who were male had no formal education, and consumed alcohol were found to be highly associated with smoking. On the other hand, individuals over the age of 75, male, residing in the eastern region, and reported to consume alcohol were linked with the use of SLT. The male tribal population, those having no formal education, those who consumed alcohol, and those belonging to the eastern region were more likely to use any form of tobacco.

The overall prevalence of tobacco observed in our study is consistent with a study from Madhya Pradesh, which found that tobacco was used by 48.1% of urban tribal people aged 40–65 years [22]. Here, it is worth noting that the prevalence of tobacco use among tribes is much higher than that of the general population (28.6%) in India, as reported by GATS-2 [5].

In comparing the prevalence of tobacco use from LASI (2017–18) and GATS-2 (2016–17), we found that in GATS-2, the prevalence of smoking was 28.6%, and SLT was 23.5% among the 45–59 years, age group. However, in our study, the prevalence of smoking was 17%, and SLT use was 31.2% among the same age group, which indicates a reduction in the use of smoking tobacco. Still, an increase in the use of SLT was observed [23].

**Table 2. Prevalence of various forms of tobacco and its univariate association with socio-demographic characteristics.**

| Socio-demographic Characteristics | Smoking | | SLT use | | Any Tobacco use | |
|---|---|---|---|---|---|---|
| | n, % (95% CI) | OR (95% CI) | n, % (95% CI) | OR (95% CI) | n, % (95% CI) | OR (95% CI) |
| **Age** | | | | | | |
| 45 to 59 years | 1014, 17.0% (16.0–18.0) | Reference | 1861, 31.2% (30.0–32.4) | Reference | 2640, 44.2% (42.9–45.5) | Reference |
| 60 to 74 years | 858, 19.6% (18.5–20.9) | 1.19 (0.97–1.47) | 1342, 30.7% (29.4–32.1) | 0.97 (0.82–1.16) | 2039, 46.7% (45.2–48.2) | 1.10 (0.93–1.31) |
| ≥75 years | 243, 23.7% (21.1–26.4) | 1.52 (1.06–2.19) | 393, 38.4% (35.4–41.4) | 1.37 (1.00–1.88) | 557, 54.4% (51.3–57.5) | 1.50 (1.13–2.01) |
| **Gender** | | | | | | |
| Male | 1790, 34.8% (33.5–36.1) | 9.67 (7.84–11.92) | 2063, 40.1% (38.7–41.4) | 2.04 (1.73–2.40) | 3417, 66.4% (65.1–67.7) | 4.77 (4.06–5.61) |
| Female | 325, 5.2% (4.7–5.8) | Reference | 1533, 24.7% (23.6–25.8) | Reference | 1819, 29.3% (28.1–30.4) | Reference |
| **Residence** | | | | | | |
| Rural | 1952, 19.5% (18.7–20.3) | 1.77 (1.21–2.59) | 3240, 32.4% (31.4–33.3) | 1.34 (1.00–1.79) | 4734, 47.3% (46.3–48.3) | 1.52 (1.14–2.04) |
| Urban | 163, 12.0% (10.4–13.9) | Reference | 356, 26.3% (24.0–28.7) | Reference | 502, 37.1% (34.5–39.7) | Reference |
| **Education** | | | | | | |
| No formal education | 1313, 17.2% (16.4–18.1) | 1.02 (0.65–1.60) | 2306, 30.3% (29.2–31.3) | 1.56 (1.02–2.38) | 3389, 44.5% (43.3–45.6) | 1.56 (1.05–2.32) |
| Up to standard VII | 556, 24.8% (23.0–26.7) | 1.63 (1.01–2.62) | 840, 37.5% (35.5–39.5) | 2.15 (1.37–3.38) | 1227, 54.8% (52.7–56.9) | 2.36 (1.54–3.62) |
| Standard VIII-X | 172, 16.2% (14.1–18.6) | 0.95 (0.56–1.60) | 354, 33.4% (30.6–36.3) | 1.80 (1.11–2.92) | 470, 44.3% (41.3–47.4) | 1.55 (0.98–2.45) |
| Higher secondary & above | 74, 16.8% (13.4–20.6) | Reference | 96, 21.8% (18.0–25.9) | Reference | 150, 33.9% (29.5–38.6) | Reference |
| **Occupation** | | | | | | |
| Currently working | 1412, 20.6% (19.7–21.6) | 1.41 (1.13–1.75) | 2391, 34.9% (33.8–36.1) | 1.47 (1.23–1.76) | 3503, 51.2% (50.0–52.4) | 1.67 (1.41–1.99) |
| Currently not working | 704, 15.6% (14.6–16.7) | Reference | 1208, 26.8% (25.5–28.1) | Reference | 1736, 38.5% (37.0–39.9) | Reference |
| **Living arrangement** | | | | | | |
| Living alone | 40, 10.6% (7.7–14.1) | 0.96 (0.52–1.75) | 142, 37.7% (32.7–42.7) | 1.37 (0.89–2.13) | 179, 47.3% (42.2–52.5) | 1.41 (0.93–2.12) |
| Living with spouse | 1764, 21.7% (20.8–22.6) | 2.26 (1.79–2.85) | 2587, 31.8% (30.7–32.8) | 1.06 (0.87–1.29) | 3951, 48.6% (47.4–49.6) | 1.48 (1.22–1.80) |
| Living without spouse | 311, 11% (9.8–12.1) | Reference | 867, 30.4% (28.7–32.2) | Reference | 1107, 38.9% (37.0–40.7) | Reference |
| **Region** | | | | | | |
| Central | 799, 23.7% (22.3–25.2) | 1.30 (0.87–1.95) | 803, 23.8% (22.4–25.3) | 3.35 (2.13–5.26) | 1513, 44.9% (43.2–46.6) | 2.28 (1.61–3.23) |
| East | 433, 14.9% (13.7–16.3) | 0.73 (0.49–1.10) | 1407, 48.6% (46.8–50.4) | 10.12 (6.55–15.64) | 1611, 55.6% (53.8–57.5) | 3.51 (2.51–4.92) |
| Northeast | 307, 22.0% (19.8–24.2) | 1.18 (0.82–1.71) | 420, 30.1% (27.7–32.5) | 4.60 (3.01–7.04) | 634, 45.4% (42.7–48.0) | 2.33 (1.69–3.21) |

*(Continued)*

**Table 2.** (Continued)

| Socio-demographic Characteristics | Smoking | | SLT use | | Any Tobacco use | |
|---|---|---|---|---|---|---|
| | n, %<br>(95% CI) | OR<br>(95% CI) | n, %<br>(95% CI) | OR<br>(95% CI) | n, %<br>(95% CI) | OR<br>(95% CI) |
| West | 281, 11.5%<br>(10.3–12.8) | 0.54 (0.34–0.85) | 802, 32.8% (30.1–34.7) | 5.24<br>(3.33–8.23) | 1037, 42.5% (40.5–44.5) | 2.07<br>(1.45–2.94) |
| South | 257, 24.1%<br>(21.6–26.8) | 1.33 (0.84–2.11) | 147, 13.8% (11.8–16.0) | 1.72<br>(1.01–2.93) | 391, 36.6% (33.8–39.7) | 1.62<br>(1.06–2.48) |
| North | 37, 19.1% (13.8–25.3) | Reference | 17, 8.5% (5.2–13.7) | Reference | 51, 26.3% (20.2–33.1) | Reference |
| **Physical Activity** | | | | | | |
| Active | 1206, 20.0% (19.0–21.0) | 1.19<br>(0.98–1.46) | 2222, 36.8% (35.6–38.1) | 1.63<br>(1.37–1.94) | 3145, 52.2% (50.9–53.4) | 1.64<br>(1.39–1.93) |
| Inactive | 909, 17.3% (16.3–18.3) | Reference | 1383, 26.3% (25.1–27.5) | Reference | 2099, 39.9% (38.6–41.3) | Reference |
| **Alcohol use** | | | | | | |
| Yes | 1198, 32.9% (31.5–34.5) | 3.60<br>(2.96–4.37) | 1909, 52.5% (50.9–54.2) | 3.90<br>(3.29–4.61) | 2753, 75.8% (74.4–77.2) | 6.48<br>(5.46–7.71) |
| No | 921, 12.0%<br>(11.3–12.8) | Reference | 1695, 22.1% (21.2–23.1) | Reference | 2494, 32.6% (31.5–33.6) | Reference |
| **MPCE Quintile** | | | | | | |
| Lowest | 619, 16.2% (15.1–17.5) | 0.70 (0.50–0.98) | 1454, 38.2% (36.6–39.7) | 2.01<br>(1.47–2.74) | 1888, 49.5% (47.9–51.1) | 1.46<br>(1.07–1.98) |
| Lower | 455, 16.0% (14.7–17.4) | 0.69 (0.49–0.97) | 899, 31.6% (29.9–33.4) | 1.51<br>(1.08–2.09) | 1276, 44.9% (43.0–46.7) | 1.21<br>(0.88–1.67) |
| Middle | 421, 21.8% (20.0–25.2) | 1.01 (0.70–1.44) | 570, 29.5% (27.5–31.6) | 1.36<br>(0.97–1.91) | 917, 47.5% (45.2–49.7) | 1.34<br>(0.97–1.86) |
| Higher | 351, 23.0%<br>(20.9–23.7) | 1.08 (0.70–1.66) | 379, 24.8% (22.7–27.1) | 1.08<br>(0.72–1.62) | 651, 42.6% (40.1–45.2) | 1.10<br>(0.75–1.61) |
| Highest | 270, 21.6% (19.3–24.0) | Reference | 294, 23.5% (21.2–25.9) | Reference | 504, 40.3% (37.5–43.0) | Reference |

Another study conducted among the Gond tribe in central India observed the prevalence of smoking to be around 22% which is in harmony with the findings of our study [24]. In contrast to the prevalence of SLT use estimated in this study, a study conducted among indigenous communities of Kerala observed the prevalence of SLT use to be around 92% [14]. A probable reason for higher SLT use among tribes could be the socio-cultural acceptance as a part of custom and perceived health benefits of SLT use compared to smoking. Nonetheless, almost half of the sample tribal population consumed nicotine, which may lead to deleterious effects on the oral cavity, respiratory, and cardiovascular systems [25].

We identified older age to be an essential predictor of tobacco use among tribal communities, which is consistent with findings of a study among a nationally representative population using GATS and GYTS that reported tobacco dependency was highest among individuals aged ≥45 years [26]. In our analysis, male participants were more likely to use tobacco than their female counterparts, similar to a study conducted among various tribes of Kerala [27]. A study conducted among the general population indicated that smoking was more prevalent among males. Conversely, smokeless tobacco usage was observed to be more common among females, indicating differences in the factors that determine tobacco usage being different among

**Table 3. Multivariable association between various forms of tobacco with socio-demographic characteristics.**

| Socio-demographic Characteristics | Smoking | SLT use | Any Tobacco use |
|---|---|---|---|
| | AOR (95% CI) | AOR (95% CI) | AOR (95% CI) |
| **Age** | | | |
| 45 to 59 years | Reference | Reference | Reference |
| 60 to 74 years | 1.10 (0.88–1.38) | 0.99 (0.82–1.20) | 1.09 (0.91–1.32) |
| ≥75 years | 1.47 (0.99–2.18) | 1.65 (1.15–2.35) | 1.76 (1.25–2.46) |
| **Gender** | | | |
| Male | 9.96 (7.76–12.77) | 1.41 (1.17–1.73) | 3.99 (3.28–4.85) |
| Female | Reference | Reference | Reference |
| **Residence** | | | |
| Rural | 1.42 (0.98–2.07) | 0.98 (0.73–1.32) | 1.01 (0.76–1.34) |
| Urban | Reference | Reference | Reference |
| **Education** | | | |
| No formal education | 2.92 (1.75–4.85) | 1.45 (0.93–2.26) | 3.09 (1.96–4.88) |
| Up to standard VII | 2.44 (1.48–4.02) | 1.78 (1.14–2.79) | 2.98 (1.88–4.72) |
| Standard VIII-X | 1.23 (0.72–2.11) | 1.57 (0.98–2.53) | 1.72 (1.07–2.76) |
| Higher secondary & above | Reference | Reference | Reference |
| **Occupation** | | | |
| Currently working | 0.84 (0.65–1.10) | 1.07 (0.86–1.33) | 1.04 (0.86–1.28) |
| Currently not working | Reference | Reference | Reference |
| **Living arrangement** | | | |
| Living alone | 0.61 (0.32–1.17) | 1.48 (0.94–2.31) | 1.31 (0.84–2.04) |
| Living with spouse | 1.18 (0.90–1.54) | 0.86 (0.70–1.07) | 0.95 (0.77–1.17) |
| Living without spouse | Reference | Reference | Reference |
| **Region** | | | |
| Central | 1.45 (0.91–2.31) | 2.38 (1.47–3.85) | 2.02 (1.37–2.99) |
| East | 0.61 (0.37–0.99) | 6.17 (3.88–9.80) | 2.52 (1.70–3.74) |
| Northeast | 1.13 (0.73–1.74) | 3.23 (2.08–5.03) | 1.94 (1.36–2.79) |
| West | 0.52 (0.31–0.85) | 4.01 (2.49–6.44) | 1.86 (1.25–2.77) |
| South | 1.20 (0.73–1.97) | 1.40 (0.81–2.41) | 1.36 (0.87–2.09) |
| North | Reference | Reference | Reference |
| **Physical Activity** | | | |
| Active | 0.97 (0.76–1.23) | 1.33 (1.09–1.63) | 1.27 (1.05–1.52) |
| Inactive | Reference | Reference | Reference |
| **Alcohol use** | | | |
| Yes | 2.09 (1.69–2.58) | 3.05 (2.54–3.66) | 4.06 (3.39–4.87) |
| No | Reference | Reference | Reference |
| **MPCE Quintile** | | | |
| Lowest | 0.51 (0.35–0.73) | 1.41 (1.04–1.92) | 0.97 (0.72–1.31) |
| Lower | 0.53 (0.36–0.77) | 1.13 (0.82–1.55) | 0.87 (0.64–1.18) |
| Middle | 0.93 (0.64–1.35) | 1.09 (0.79–1.52) | 1.11 (0.81–1.52) |
| Higher | 0.93 (0.62–1.37) | 0.99 (0.67–1.46) | 0.96 (0.67–1.38) |
| Highest | Reference | Reference | Reference |

different groups [23]. Additionally, the study found a correlation between tobacco usage and lower levels of educational attainment. Smoking tobacco was more common among those who did not have formal education, whereas participants who completed primary education were

more frequent users of the SLT. These findings are similar to a study conducted among urban tribal people in Madhya Pradesh, which reported maximum tobacco consumption among participants with lesser years of schooling [22].

Our study revealed that individuals who engaged in physical activity were more prone to consume SLT and any tobacco product in comparison to those who didn't. However, Jeon et al. (2021) discovered that there were marked differences in physical fitness between smokers and non-smokers, with the latter having a higher level of physical capability [28]. We found smoking was insignificantly associated with physical activity which was inconsistent with the results of a study conducted by Cram et al. (2014) found that individuals who smoked and were less physically active tended to have poor health outcomes in the population [29].

Tribal individuals belonging to higher MPCE quintiles were found to have a greater risk of smoking. However, a review suggested smoking to be higher in the medium MPCE quintiles group of the general population [13]. It also reported that SLT use in India was more among the lowest MPCE quintile class [13]. We observed tobacco use was associated with the simultaneous use of alcohol among our study participants [30]. A recent meta-analysis reported that the synergistic consumption of any form of tobacco with alcohol is significantly associated with a higher risk of oral squamous cell carcinoma [31]. NFHS-5 (2019–2021) a nationally representative survey, suggests that tobacco prevalence was higher in Northeastern states. Our present study also observed a higher odds of SLT and any form of tobacco use in the eastern region. NFHS-5 showed a decline in tobacco use prevalence among most of the states except Bihar, Himachal Pradesh, and Mizoram, where still an uptrend can be seen [32]. We also estimated a higher tobacco burden in states such as Bihar, West Bengal, Jharkhand, and Odisha in the present study. A study conducted by Yadav et al. (2020) to evaluate the burden of SLT use indicates that the prevalence in eastern states, Bihar (23.5%), Jharkhand (35.4%), and Odisha (42.9%) was higher than the national average (21.4%), whereas in West Bengal (20.1%) it is less than the national average [33]. This could be due to various factors including family environment (as the child learns from immediate family members), and social traditions, both containing the elements that promoted tobacco use [34]. The peer pressure, easy access to tobacco and *gutkha* at shops nearby, and the absence of opposition from the parents, or other family members likely contributed to tobacco use [34–36]. Additionally, tobacco is used as a mean to concentrate on work [37]. A study conducted by Nanda et al. among Donghria Kondhs tribes in Odisha showed that tobacco is used to reduce abdominal pain during menstruation [38]. These are some of the driving factors that influence tobacco use among tribals. The regional differences emphasize further strengthening of tobacco control programs and policies with a focus on the eastern tribal populations.

Even though tobacco is banned in most states, the prevalence of tobacco consumption is still high in Bihar and Odisha, as evidenced by our findings. The Jharkhand Government, in a recent order, has banned the consumption of any form of tobacco products for all State Government employees from 01 April 2021. Such initiatives should be taken in all other states for tobacco control.

Several existing tobacco control programs and policies, such as the National Tobacco Control Programme (NTCP) and The Cigarettes and Other Tobacco Products (Prohibition of Advertisement and Regulation of Trade and Commerce, Production, Supply, and Distribution) Act, COTPA, 2003, have been effective in reducing the burden of tobacco in India [39]. SLT taxation as a policy imperative remains an under-researched area and needs further attention from researchers and policymakers for effective outcomes [40]. Still, a wide gap exists between the burden of tobacco consumption among tribes and the general population. Our study reflects the prevalence of tribal tobacco use is higher than the average national use,

which calls for urgent action in the tribal-dominated regions [5]. The existing programs should be tailored so that they are culturally acceptable and linguistically understandable among these groups, which can help create a positive effect.

## Implications

Our study suggests that the prevalence of tobacco use was high among those who had no formal education. So, culturally targeted messages are a more effective way to testify positive effects on individual health status and control tobacco among indigenous populations [41]. Culturally appropriate and acceptable behavioural change communication (BCC) is required for this group. Folk dances, puppet shows, and other such traditions can be employed for better acceptability and interest. Socio-cultural and religious events of particular tribal groups can be used to communicate anti-tobacco messages. Anti-tobacco campaigns should use messages in the local language about the specific tribal group for their better understanding and arouse interest in the subject matter. In addition, the tribal youths who are literate, local, and share a standard dialect can actively participate in the propagation of tobacco-related hazards and associated health risks. Widespread sensitization attempts to resolve the gap regarding tribal focused plans and leveraging government programs in the context of tribal health may settle the tobacco problem. Although tobacco consumption is high irrespective of gender, males are still at a higher risk of tobacco use, making them the priority target group. The Eastern region needs a strengthened program owing to the higher burden.

## Strength and limitations

LASI provides an extensive and nationally representative dataset and an adequate sample size. It provides valuable data on the consumption of various forms of tobacco such as smoking, SLT, and any tobacco use. It establishes a strong association between the prevalence of tobacco and various socio-demographic factors. However, we considered the self-reported cases of different forms of tobacco use, which may lead to underestimation (misclassification bias, recall bias) of the actual burden of tobacco. The cross-sectional design hampered our abilities to draw causal conclusions and investigate longitudinal relationships over time. Additionally, the sampling weights utilized in this study are for the general population.

## Conclusion

This study highlights the high burden of tobacco use along with its social determinants, which can help in tailoring the tobacco control program among tribes in India. Tobacco control policies should target males, rural residents, individuals without formal education, and alcoholics. Regional variations should be managed by adopting good practices of regions with higher tobacco control. Tobacco control measures should be prioritized in high-burden states such as Bihar, Jharkhand, Odisha, and West Bengal. Future studies should explore the behavioural and social linkages to tobacco use among tribals.

## Supporting information

**S1 Table. Strengthening the Reporting of Observational Studies in Epidemiology (STROBE) guideline to report this study.**
(DOC)

**S2 Table. Additional information regarding the ethical, cultural, and scientific considerations specific to inclusivity in global research is included in the Supporting Information.**
(DOCX)

**S3 Table. State wise prevalence of different forms of tobacco in India among tribes.** (DOCX)

## Acknowledgments

The authors are grateful to the Longitudinal Ageing Study in India (LASI) for assembling and publishing accurate, nationally representative data on a range of health, biomarkers, and healthcare utilization indicators for the population in the age range of 45 years and older. The authors are also grateful to LASI's project partners, the International Institute for Population Sciences (IIPS), Harvard T. H. Chan School of Public Health (HSPH), and the University of Southern California (USC).

## Author Contributions

**Conceptualization:** Shishirendu Ghosal, Abhinav Sinha, Srikanta Kanungo, Sanghamitra Pati.

**Data curation:** Jogesh Murmu, Srikanta Kanungo, Sanghamitra Pati.

**Formal analysis:** Jogesh Murmu, Ritik Agrawal, Sayantani Manna, Sweta Pattnaik, Shishirendu Ghosal, Abhinav Sinha, Ardhendu Sekhar Acharya.

**Investigation:** Jogesh Murmu, Ritik Agrawal, Sayantani Manna, Sweta Pattnaik, Shishirendu Ghosal, Abhinav Sinha, Ardhendu Sekhar Acharya, Sanghamitra Pati.

**Methodology:** Jogesh Murmu, Ritik Agrawal, Sayantani Manna, Sweta Pattnaik, Shishirendu Ghosal, Abhinav Sinha, Ardhendu Sekhar Acharya, Srikanta Kanungo, Sanghamitra Pati.

**Project administration:** Srikanta Kanungo, Sanghamitra Pati.

**Resources:** Srikanta Kanungo, Sanghamitra Pati.

**Software:** Srikanta Kanungo, Sanghamitra Pati.

**Supervision:** Srikanta Kanungo, Sanghamitra Pati.

**Validation:** Shishirendu Ghosal, Ardhendu Sekhar Acharya, Srikanta Kanungo, Sanghamitra Pati.

**Visualization:** Jogesh Murmu, Ritik Agrawal, Sayantani Manna, Sweta Pattnaik, Abhinav Sinha.

**Writing – original draft:** Jogesh Murmu, Ritik Agrawal, Sayantani Manna, Sweta Pattnaik, Shishirendu Ghosal, Abhinav Sinha.

**Writing – review & editing:** Ardhendu Sekhar Acharya, Srikanta Kanungo, Sanghamitra Pati.

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
