## [Decision Letter · Decision Letter 0]

28 Jun 2022

PONE-D-22-13582Social determinants of tobacco use among tribal communities in India: evidence on ethnicity and tobacco use from LASI, wave-1PLOS ONE

Dear Dr. Pati,

Thank you for submitting your manuscript to PLOS ONE. After careful consideration, we feel that it has merit but does not fully meet PLOS ONE’s publication criteria as it currently stands. Therefore, we invite you to submit a revised version of the manuscript that addresses the points raised during the review process.

We look forward to receiving your revised manuscript.

Kind regards,

Jayanta Kumar Bora,PhD

Academic Editor

PLOS ONE

Journal Requirements:

4. Please ensure that you refer to Figure 1 in your text as, if accepted, production will need this reference to link the reader to the figure.

5. We note that Figure 3 in your submission contain map images which may be copyrighted. All PLOS content is published under the Creative Commons Attribution License (CC BY 4.0), which means that the manuscript, images, and Supporting Information files will be freely available online, and any third party is permitted to access, download, copy, distribute, and use these materials in any way, even commercially, with proper attribution. For these reasons, we cannot publish previously copyrighted maps or satellite images created using proprietary data, such as Google software (Google Maps, Street View, and Earth). For more information, see our copyright guidelines: http://journals.plos.org/plosone/s/licenses-and-copyright.

a. You may seek permission from the original copyright holder of Figure 3 to publish the content specifically under the CC BY 4.0 license.  

Reviewers' comments:

Reviewer's Responses to Questions

**Comments to the Author**

1. Is the manuscript technically sound, and do the data support the conclusions?

Reviewer #1: Yes

Reviewer #2: Partly

Reviewer #3: Yes

2. Has the statistical analysis been performed appropriately and rigorously? 

Reviewer #1: Yes

Reviewer #2: No

Reviewer #3: Yes

3. Have the authors made all data underlying the findings in their manuscript fully available?

Reviewer #1: Yes

Reviewer #2: No

Reviewer #3: Yes

4. Is the manuscript presented in an intelligible fashion and written in standard English?

Reviewer #1: Yes

Reviewer #2: Yes

Reviewer #3: Yes

5. Review Comments to the Author

Reviewer #1: There are few things to be corrected within the already written paper, which were also highlighted in supporting reviewed file.

The authors must go through folloing points:

1. Go through some already published plosone paper's and make proper citation.

2. For some words use synonyms or different words.

3. Mention why you have categorised age as 14 years difference, the answer my be supported with previous literature.

4. If the authors has taken "living arrangent variable" for 'living with partner' and 'living without partner', then how the authors manage 'living alone category'. Also in Independent variable/covariates section, mention the name of "variable taken from LASI", and how you have recoded the variables for your study.

5. Go through some already published plosone paper's and make tables properly. Also, use Inactive instead of not active, its sound better in research article.

6. In table 2, authors have make hapazard table. Please try to take the same category in reference among all parameters. Its better to re-do the table 2, specially for variables like- age, education, life-partner, religion, and asset quintile. If not agreed for re-doing then give valid reason for keeping such vigorous category referencing in methodilogy section.

7. Asset quintile category should be named in a similar fashion, keep same type of naming for all five categories. Don't use 2,3,4 suddenly in between most deprived and most affluent. Authors may use (extremely, less, medium, high, very high) type of categorization.

8. Do not use 'our participants'. Use correct english. Authors may use 'study participants'.

There are few things that are required to do for this paper,

1. Most of the literature cited were taken from less prevalent states. (Example- Madhya Pradesh, & Kerala). But north-eastern are highly tobacco prevalent states. Please go through some papers on north-eastern states, there are plenty of works available for north eastern states.

2. In conclusion writing a line like (Regional variations should be managed by adopting good practices of regions with higher tobacco control) is not justified. Authors must name the states where government intervention is required.

3. Authors have perfecly done a regional analysis. But naming the states within that region is necessary. For example there is high prevalence of tobacco in north eastern states, and Assam government banned tobacco products on 22 November 2019. Then Tripura government can take similar steps, is a conclusion. Authors may write conclusion section in such a way.

4. Last but not least, authors should concentrate more on high tribal population states like Madhya Pradesh, Orissa, Maharashtra, Rajasthan, & Chattisgarh. Alongwith higher percentage of Tribal population states like Arunachal Pradesh, Nagaland, Mizoram, Meghalaya, & Manipur.

Reviewer #2: In this manuscript authors have made an attempt to estimate the prevalence of tobacco use among tribal population aged 45 years or above in India based on secondary data-LASI. Efforts shown by the authors are good but there some major concerns in this current version of the manuscript listed below:

1. The literature search appears less robust and authors have missed on many recent publications. The high prevalence of tobacco use among this socially vulnerable group has been already established in the last literature and information provided here is not adding anything new to the existing knowledge. Therefore, rationale to conduct this study on tobacco use among aged tribal population needs a stronger rationale.

2. The citations need to be updated and should follow a specific format as per the journal guideline. There are also some repetitions or in a format difficult to understand say for example 18. 19

3. In methodology section, the analysis part seems incomplete. Even for regression analysis, what covariates were adjusted were not clear. The association between alcohol consumption and tobacco use might be misleading unless controlling covariates are clearly mentioned. It is also not clear how did they adjust for sample weights. The presentations in the the table appear much cluttered and hard to comprehend.

4. Apart from misclassification error, there are possibilities of other potential sources of biases which authors didn't mention and also how did authors try to adjust for such biases is also not mentioned

5. The overall writing style and presentations needs to be more comprehensive

Reviewer #3: The paper is trying to address an important question on tobacco use behaviour of the tribal population in India and understand their determinants within this group. However, I do have some major comments to improve this paper.

1. This paper is based on LASI data which is nationally and state representative. However, since this study is conducted on a specific set of population, any estimation for ST population at the state level may not be appropriate, as the total sample of ST population in this data is 11,365 tribal individuals aged ≥ 45 years. Segregating this data at the state level will lead to very few cases in smaller states. Therefore, I would recommend Authors to remove Figure 3 completely from this paper. If they want they can provide a table of prevalence of tobacco use in bigger states only in the form of a table in place of Figure 3.

2. I also fail to understand the reason for taking difference reference group each time for wealth quintile for tobacco, Smokeless and smoking outcome. Similarly, west region was used as reference for smoking and North for Smokeless and tobacco use. Can the Authors provide their reason for changing references in these models for the same socio demographic characteristics, or they should make these reference consistent and re do the logistic regression analysis, otherwise it may be difficult to compare the results for different form of tobacco use and their association with these characteristics.

3. Are there any evidence which suggest that physical activity leads to higher/lower tobacco use. I did find literature the other way around. Authors should provide some appropriate references to substantiate their choice of physical activity as one of the variable.

Jeon, H. G., Kim, G., Jeong, H. S., & So, W. Y. (2021, February). Association between Cigarette Smoking and Physical Fitness Level of Korean Adults and the Elderly. In Healthcare (Vol. 9, No. 2, p. 185). MDPI.

Conway, T. L., & Cronan, T. A. (1992). Smoking, exercise, and physical fitness. Preventive medicine, 21(6), 723-734.

4. GATS-2 has collected information about the tobacco consumption and provide caste wise data as well. Why the authors have chosen LASI data for this specific study need to be mentioned clearly. For example, GATS collected selective socioeconomic characteristics data while LASI provides a detail information on this front.

5. Authors can also provide some comparison of GATS-2 and LASI tobacco use data for ST population in the discussion.

6. PLOS authors have the option to publish the peer review history of their article (what does this mean?). If published, this will include your full peer review and any attached files.

Reviewer #1: **Yes: **Tushar Dakua

Reviewer #2: No

Reviewer #3: **Yes: **Akansha Singh

---

## [Author Response · Author response to Decision Letter 0]

22 Jul 2022

Point-by-Point Response Reviewers’ Comments

We would like to thank the Editor and reviewers for their thoughtful comments and efforts towards improving our manuscript. The suggestions offered have been extremely helpful in revising the manuscript. We have incorporated all the changes suggested. Please find the response to all the comments highlighted in a point-by-point basis. We hope the revised manuscript is appropriate for publication in the journal. But we are open to further revisions if required. 

Reviewer #1: There are few things to be corrected within the already written paper, which were also highlighted in supporting reviewed file.

The authors must go through following points:

1. Go through some already published plos one paper's and make proper citation.

Authors’ reply- Thank you for your suggestions we have gone through the literature and changed our citations accordingly.

2. For some words use synonyms or different words.

Authors’ reply- We have tried to improve the entire manuscript by editing it and we have added synonyms also as suggested by the reviewer.

3. Mention why you have categorized age as 14 years difference, the answer may be supported by previous literature.

Authors’ reply- We have categorized age in line with the previous literatures which have now been cited in the manuscript.

4. If the authors have taken "living arrangement variable" for 'living with partner' and 'living without partner', then how the authors manage 'living alone category'. Also, in independent variable/covariates section, mention the name of "variable taken from LASI", and how you have recoded the variables for your study.

Authors’ reply- Initially we had taken dm005 (current marital status) as our partner status variable. But as suggested by the reviewer, we have taken the new variable from the LASI dataset which is the “living arrangement variable” to include the category of living alone.

5. Go through some already published plos one paper's and make tables properly. Also, use Inactive instead of not active, its sound better in research article.

Authors’ reply- As per the suggestions of the reviewer we have changed the tables according to plos-one guidelines and included standard fonts and standard size. Additionally, for physical activity we have changed not active to inactive.

6. In table 2, authors have make a haphazard table. Please try to take the same category in reference among all parameters. Its better to re-do table 2, specially for variables like- age, education, life-partner, religion, and asset quintile. If not agreed for re-doing then give valid reason for keeping such vigorous category referencing in the methodology section.

Authors’ reply- Previously we had taken different reference groups as mentioned by the reviewer. However, as per the suggestions we have redone the table-2 (univariate regression analysis) and we have kept uniform reference groups of all the models.

7. Asset quintile category should be named in a similar fashion, keep same type of naming for all five categories. Don't use 2,3,4 suddenly in between most deprived and most affluent. Authors may use (extremely, less, medium, high, very high) type of categorization.

Authors’ reply- Thank you for the suggestion. We have changed the names of five categories of asset quintile to poorest, poorer, middle, richer, and richest.

8. Do not use 'our participants'. Use correct English. Authors may use 'study participants'.

Authors’ reply- Thank you for pointing this out. We have now revised it with study participants in our manuscript.

There are few things that are required to do for this paper,

1. Most of the literature cited were taken from less prevalent states. (Example- Madhya Pradesh, & Kerala). But north-eastern are highly tobacco prevalent states. Please go through some papers on north-eastern states, there are plenty of works available for north eastern states.

Authors’ reply- We agree with the reviewer’s suggestions. However, most of the literature from north eastern region mainly focussed either on general population or on tribal adolescents with almost no study in context to older tribal adults which makes our study incomparable with them. This has forced us to not cite those studies in our manuscript and also this is the novelty of our study that it focuses on an age group in which less attention has been paid till now as most studies focus on young adults.

2. In conclusion writing a line like (Regional variations should be managed by adopting good practices of regions with higher tobacco control) is not justified. Authors must name the states where government intervention is required.

Authors’ reply- Thank you for your suggestions. This is an important point that we have missed and it is worth noting the name of the states where the burden of tobacco use is high and we have addressed this point in the conclusion under the regional variations paragraph.

3. Authors have perfectly done a regional analysis. But naming the states within that region is necessary. For example, there is high prevalence of tobacco in north eastern states, and Assam government banned tobacco products on 22 November 2019. Then Tripura government can take similar steps, is a conclusion. Authors may write conclusion section in such a way.

Authors’ reply- The reviewer has raised a very valid point and we have now mentioned the names of the states. Additionally, in the discussion section we have discussed on the similar lines as suggested by the reviewer.

4. Last but not least, authors should concentrate more on high tribal population states like Madhya Pradesh, Orissa, Maharashtra, Rajasthan, & Chhattisgarh. Along with a higher percentage of Tribal population states like Arunachal Pradesh, Nagaland, Mizoram, Meghalaya, & Manipur. 

Authors’ reply- Thank you for the suggestion. We have now focussed on high tribal populated states and mentioned it in the discussion section.

We again thank the reviewer for their valuable time and inputs.

Reviewer #2: In this manuscript, authors have made an attempt to estimate the prevalence of tobacco use among tribal population aged 45 years or above in India based on secondary data-LASI. Efforts shown by the authors are good but there some major concerns in this current version of the manuscript listed below:

1. The literature search appears less robust and authors have missed on many recent publications. The high prevalence of tobacco use among this socially vulnerable group has been already established in the last literature and information provided here is not adding anything new to the existing knowledge. Therefore, rationale to conduct this study on tobacco use among aged tribal population needs a stronger rationale.

Authors’ reply- We appreciate the observations of reviewer. However, we affirm that the available literature among the tribal focuses mainly on younger age groups with a scarcity of data among older adult. We generated evidence using a nationally representative data of older tribal adults which is novel and a strength of this study. 

2. The citations need to be updated and should follow a specific format as per the journal guideline. There are also some repetitions or in a format difficult to understand say for example 18. 19

Authors’ reply- We have gone through the guidelines of plos one for citation of articles and have modified the references accordingly.

3. In the methodology section, the analysis part seems incomplete. Even for regression analysis, what covariates were adjusted were not clear. The association between alcohol consumption and tobacco use might be misleading unless controlling covariates are clearly mentioned. It is also not clear how did they adjust for sample weights. The presentations on the table appear much cluttered and hard to comprehend.

Authors’ reply- For multivariable regression analysis we have adjusted for all the socio-demographic variables. WE have now mentioned this in the statistical analysis portion of the methodology. Sampling weights have been utilized in entire analysis to compensate for the complex survey design which has now been mentioned in the manuscript for better understanding of the readers.

4. Apart from misclassification error, there are possibilities of other potential sources of biases which the authors didn't mention and also how did authors try to adjust for such biases is also not mentioned

Authors’ reply- Thank you for your suggestion. We have now addressed other potential biases also. 

5. The overall writing style and presentations need to be more comprehensive

Authors’ reply- As per your suggestion we have now modified the entire manuscript to check for grammar, punctuations in order to enhance writing and presentations style.

We again thank the reviewer for their valuable time and inputs.

Reviewer #3: The paper is trying to address an important question on tobacco use behaviour of the tribal population in India and understand their determinants within this group. However, I do have some major comments to improve this paper.

1. This paper is based on LASI data which is nationally and state representative. However, since this study is conducted on a specific set of population, any estimation for ST population at the state level may not be appropriate, as the total sample of ST population in this data is 11,365 tribal individuals aged ≥ 45 years. Segregating this data at the state level will lead to very few cases in smaller states. Therefore, I would recommend Authors to remove Figure 3 completely from this paper. If they want, they can provide a table of prevalence of tobacco use in bigger states only in the form of a table in place of Figure 3.

Authors’ reply- As per your suggestions, we have removed the figure-3 (State-wise prevalence of various forms of tobacco among tribals in India). Instead of that, we have added the state-wise prevalence of various forms of tobacco in the supplementary file and also added the figure of top 4 tribal populated states according to census-2011 population.

2. I also fail to understand the reason for taking difference reference group each time for wealth quintile for tobacco, Smokeless and smoking outcome. Similarly, west region was used as reference for smoking and North for Smokeless and tobacco use. Can the Authors provide their reason for changing references in these models for the same socio demographic characteristics, or they should make these reference consistent and re do the logistic regression analysis, otherwise it may be difficult to compare the results for different form of tobacco use and their association with these characteristics.

Authors’ reply- Thank you for this valuable suggestion. We have changed the reference groups of the variables where the reference seems to be inconsistent like age, asset quintile, education age, region, living arrangement variables. And we have redone table-2 and kept the reference group same as suggested by the reviewer.

3. Are there any evidence which suggest that physical activity leads to higher/lower tobacco use. I did find literature the other way around. Authors should provide some appropriate references to substantiate their choice of physical activity as one of the variables.

Jeon, H. G., Kim, G., Jeong, H. S., & So, W. Y. (2021, February). Association between Cigarette Smoking and Physical Fitness Level of Korean Adults and the Elderly. In Healthcare (Vol. 9, No. 2, p. 185). MDPI.

Conway, T. L., & Cronan, T. A. (1992). Smoking, exercise, and physical fitness. Preventive medicine, 21(6), 723-734.

Authors’ reply- The choice of physical activity as one of the variables as most of the tribals are involved in vigorous activities and it was mentioned in detail in the methodology section. However, the suggested literature have been used to compare our findings in the discussion section.

4. GATS-2 has collected information about tobacco consumption and provides caste-wise data as well. Why the authors have chosen LASI data for this specific study needs to be mentioned clearly. For example, GATS collected selective socioeconomic characteristics data while LASI provides a detailed information on this front.

Authors’ reply- We have given justification for selection of LASI data for our analysis.

5. Authors can also provide some comparison of GATS-2 and LASI tobacco use data for ST population in the discussion.

Authors’ reply- We have addressed the comparison in the discussion section and compared the GATS-2 and LASI tobacco use data for the scheduled tribes population.

We again thank the reviewer for their valuable time and inputs.

---

## [Decision Letter · Decision Letter 1]

28 Nov 2022

PONE-D-22-13582R1Social determinants of tobacco use among tribal communities in India: evidence on ethnicity and tobacco use from LASI, wave-1PLOS ONE

Dear Dr. Pati,

Thank you for submitting your manuscript to PLOS ONE. After careful consideration, we feel that it has merit but does not fully meet PLOS ONE’s publication criteria as it currently stands. Therefore, we invite you to submit a revised version of the manuscript that addresses the points raised during the review process. =============================

ACADEMIC EDITOR:

Dear Authors,

Please address the following comments:

1) In the title, “Social determinants of tobacco use among tribal communities in India: evidence on ethnicity and tobacco use from LASI, wave-1”, the term like ‘tobacco use’ is repetitive. Again, the term tribal has already been mentioned in the title, the term ‘ethnicity’ may not be required. The title may be simplified to ‘Social determinants of tobacco use among tribal communities in India: evidence from the first wave of LASI (full form??)’.

2) Why were smokers and users of tobacco in dual for clubbed into smokers? They are one more dangerous category. The study might have a greater weightage if the severity of the tobacco consumption is considered like only smokeless/other form users, only smokers and both smokers and non-smokers.

4) The sampling weights given in LASI are meant for the general population and may not be applicable to the tribal population.

5) P10 L212: ‘For this analysis, 11,365 scheduled tribes aged 45 years and above were included.’ I think you mean scheduled tribe participants here?

6) It should be clearly mentioned in the methodology that the wealth quintiles have been generated from the general population, not from the tribal population.

7) Table 2 headings are given as n (%) but data is reflected in n, % (CI). The variable names may be aligned to left (Age, Gender etc). Why are one reference category at the top and the other at the bottom?

8) Why do tribal groups from eastern regions have the highest tobacco consumption? This may be reflected from the existing literature. Does it reflect socio-cultural upbringing of any major tribal group/s?

9) Proper referencing should be done for some of the references like “National_Fact_Sheet_of_fourth_round_of_Global_Youth_Tobacco_Survey_GYTS- 4.pdf.”

10) Please correct the spelling of state names like Maharashtra and Rajasthan in Figure 3.

11) Manuscripts may be carefully edited for scientific language and grammatical errors. Please include the following items when submitting your revised manuscript:A rebuttal letter that responds to each point raised by the academic editor and reviewer(s). You should upload this letter as a separate file labeled 'Response to Reviewers'.A marked-up copy of your manuscript that highlights changes made to the original version. You should upload this as a separate file labeled 'Revised Manuscript with Track Changes'.An unmarked version of your revised paper without tracked changes. You should upload this as a separate file labeled 'Manuscript'.

We look forward to receiving your revised manuscript.

Kind regards,

Mohsen Abbasi-Kangevari

Academic Editor

PLOS ONE

Reviewers' comments:

Reviewer's Responses to Questions

**Comments to the Author**

1. If the authors have adequately addressed your comments raised in a previous round of review and you feel that this manuscript is now acceptable for publication, you may indicate that here to bypass the “Comments to the Author” section, enter your conflict of interest statement in the “Confidential to Editor” section, and submit your "Accept" recommendation.

Reviewer #1: All comments have been addressed

Reviewer #4: (No Response)

2. Is the manuscript technically sound, and do the data support the conclusions?

Reviewer #1: Yes

Reviewer #4: Yes

3. Has the statistical analysis been performed appropriately and rigorously? 

Reviewer #1: Yes

Reviewer #4: Yes

4. Have the authors made all data underlying the findings in their manuscript fully available?

Reviewer #1: Yes

Reviewer #4: Yes

5. Is the manuscript presented in an intelligible fashion and written in standard English?

Reviewer #1: Yes

Reviewer #4: Yes

6. Review Comments to the Author

Reviewer #1: I have reviewed the paper comprehensively. I found that authors have addressed the issues, which I have raised earlier.

Reviewer #4: This present study deals with an important aspect of social determinants of tobacco use among tribals in India. The paper is revised by the authors after the first round of review. However, I have the following comments on the paper:

1) In the title, “Social determinants of tobacco use among tribal communities in India: evidence on ethnicity and tobacco use from LASI, wave-1”, the term like ‘tobacco use’ is repetitive. Again, the term tribal has already been mentioned in the title, the term ‘ethnicity’ may not be required. The title may be simplified to ‘Social determinants of tobacco use among tribal communities in India: evidence from the first wave of LASI (full form??)’.

2) Why were smokers and users of tobacco in dual for clubbed into smokers? They are one more dangerous category. The study might have a greater weightage if the severity of the tobacco consumption is considered like only smokeless/other form users, only smokers and both smokers and non-smokers.

4) The sampling weights given in LASI are meant for the general population and may not be applicable to the tribal population.

5) P10 L212: ‘For this analysis, 11,365 scheduled tribes aged 45 years and above were included.’ I think you mean scheduled tribe participants here?

6) It should be clearly mentioned in the methodology that the wealth quintiles have been generated from the general population, not from the tribal population.

7) Table 2 headings are given as n (%) but data is reflected in n, % (CI). The variable names may be aligned to left (Age, Gender etc). Why are one reference category at the top and the other at the bottom?

8) Why do tribal groups from eastern regions have the highest tobacco consumption? This may be reflected from the existing literature. Does it reflect socio-cultural upbringing of any major tribal group/s?

9) Proper referencing should be done for some of the references like “National_Fact_Sheet_of_fourth_round_of_Global_Youth_Tobacco_Survey_GYTS- 4.pdf.”

10) Please correct the spelling of state names like Maharashtra and Rajasthan in Figure 3.

11) Manuscripts may be carefully edited for scientific language and grammatical errors.

7. PLOS authors have the option to publish the peer review history of their article (what does this mean?). If published, this will include your full peer review and any attached files.

Reviewer #1: **Yes: **Tushar Dakua

Reviewer #4: No

---

## [Author Response · Author response to Decision Letter 1]

6 Dec 2022

Point-by-Point Response Reviewers’ Comments

We would like to thank the Editor and reviewers for their thoughtful comments and efforts towards improving our manuscript. The suggestions offered have been extremely helpful in revising the manuscript. We have incorporated all the changes suggested. Please find the response to all the comments highlighted in a point-by-point basis. We hope the revised manuscript is appropriate for publication in the journal. But we are open to further revisions if required. 

Reviewer #4: This present study deals with an important aspect of social determinants of tobacco use among tribals in India. The paper is revised by the authors after the first round of review. However, I have the following comments on the paper:

1. In the title, “Social determinants of tobacco use among tribal communities in India: evidence on ethnicity and tobacco use from LASI, wave-1”, the term like ‘tobacco use’ is repetitive. Again, the term tribal has already been mentioned in the title, the term ‘ethnicity’ may not be required. The title may be simplified to ‘Social determinants of tobacco use among tribal communities in India: evidence from the first wave of LASI (full form??)’.

Authors’ reply- Thank you for your suggestion. We have changed the title of the study as suggested.

2. Why were smokers and users of tobacco in dual for clubbed into smokers? They are one more dangerous category. The study might have a greater weightage if the severity of the tobacco consumption is considered like only smokeless/other form users, only smokers and both smokers and non-smokers.

Authors’ reply- when we created the binary variable "smokers" as an outcome variable, both only smokers and dual users were merged; the sub-groups under smokers were smokers and non-smokers (includes smokeless tobacco users but don't use smoking tobacco and those who don't use any kind of tobacco products). Similarly, when we created another binary variable as "smokeless tobacco users", the subgroups under this variable were SLT users (only SLT users + dual users, as they used SLT products also) and non-SLT users (included those who didn't use tobacco of any kind & those who were consumer of smoking tobacco only). Any tobacco group included all kind of tobacco users (only smokers+ only SLT users + dual users) irrespective of type of products they were using. We conducted three separate logistic regression model for smokers, smokeless tobacco users and any tobacco users to avoid collinearity.

4. The sampling weights given in LASI are meant for the general population and may not be applicable to the tribal population.

Authors’ reply- We agree with the reviewer. However, studies such as “Puri P, Pati S. Exploring the Linkages Between Non-Communicable Disease Multimorbidity, Health Care Utilization and Expenditure Among Aboriginal Older Adult Population in India. International Journal of Public Health. 2022:5.” have utilized survey weights in analysing LASI data for tribal population. We feel weighted analysis can better present the exact scenario; hence we have used weighted analysis. Nonetheless, following reviewers’ suggestion in methods section we have mentioned this as one of the limitations of the study.

5. P10 L212: ‘For this analysis, 11,365 scheduled tribes aged 45 years and above were included.’ I think you mean scheduled tribe participants here? 

Authors’ reply- We have added the word “participants” as suggested by the reviewer.

6. It should be clearly mentioned in the methodology that the wealth quintiles have been generated from the general population, not from the tribal population. 

Authors’ reply- As suggested by the reviewer, we have mentioned that the wealth quintiles are based on monthly per capita expenditure of the household from the general population, not from the tribal population.

7. Table 2 headings are given as n (%) but data is reflected in n, % (CI). The variable names may be aligned to left (Age, Gender etc). Why are one reference category at the top and the other at the bottom?

Authors’ reply- As per the suggestions by the reviewer we have changed the headings and subheadings in table-2 and wherever it is required in our study. We have placed all the reference groups in bottom but in age category the 1st category (i.e., 45-59 years) was taken as reference, hence it remained at the top of the table.

8.Why do tribal groups from eastern regions have the highest tobacco consumption? This may be reflected from the existing literature. Does it reflect socio-cultural upbringing of any major tribal group/s?

Authors’ reply- We have now added this in the discussion section. 

9. Proper referencing should be done for some of the references like “National_Fact_Sheet_of_fourth_round_of_Global_Youth_Tobacco_Survey_GYTS- 4.pdf.”

Authors’ reply- Thank you for pointing this out. All references have been revised and re-checked. 

10. Please correct the spelling of state names like Maharashtra and Rajasthan in Figure-3

Authors’ reply- As per the reviewer’s suggestion all the spelling has been corrected in the figure-3

11. Manuscripts may be carefully edited for scientific language and grammatical errors.

Authors’ reply- Manuscript has been checked for grammatical error and corrected accordingly.

We again thank the reviewer for their valuable time and inputs. ________________________________________

---

## [Decision Letter · Decision Letter 2]

1 Feb 2023

PONE-D-22-13582R2Social determinants of tobacco use among tribal communities in India: evidence from the first wave of Longitudinal Aging study of IndiaPLOS ONE

Dear Dr. Pati,

Thank you for submitting your manuscript to PLOS ONE. After careful consideration, we feel that it has merit but does not fully meet PLOS ONE’s publication criteria as it currently stands. Therefore, we invite you to submit a revised version of the manuscript that addresses the following points: Academic Editor's Comments:

1. The revised manuscript still needs language editing in some of its sections. Please follow some of the editing done in the attached copy of the manuscript (with track change).

2. In line 185-187: Authors have used the sentence “Wealth quintiles (poorest, poorer, middle, richer, richest) were based on the monthly per capita expenditure of the general population.” LASI data does not provide wealth quintile; it is MPCE quintile and they can’t be used interchangeably. So, replace the wealth quintile with the MPCE quintile, wherever used. Authors can avoid the limitation of using the indicator calculated based on the general population by calculating the MPCE quintile for their sample (tribal population) as well. This can be easily done using the expenditure details on food and non-food items including spending on health, education, utilities etc. given in the household dataset.

3. Follow the other comments and suggestions mentioned in the attached edited copy of the manuscript.

We look forward to receiving your revised manuscript.

Kind regards,

Chandan Kumar, Ph.D.

Academic Editor

PLOS ONE
---

## [Author Response · Author response to Decision Letter 2]

3 Feb 2023

Response to Reviewers

We would like to thank the Editor and reviewers for their thoughtful comments and efforts toward improving our manuscript. The suggestions offered have been extremely helpful in revising the manuscript. We have incorporated all the changes suggested. Please find the response to all the comments highlighted on a point-by-point basis. We hope the revised manuscript is appropriate for publication in the journal. But we are open to further revisions if required. 

Academic Editor's Comments:

1. The revised manuscript still needs language editing in some of its sections. Please follow some of the editing done in the attached copy of the manuscript (with track change).

Author’s Response: Thank you for your suggestion. We have changed the revised manuscript with language editing as suggested by the editors. In lines 259-260: as suggested by the editor, we have replaced smokeless with SLT and followed with throughout the manuscript.

2. In line 185-187: Authors have used the sentence “Wealth quintiles (poorest, poorer, middle, richer, richest) were based on the monthly per capita expenditure of the general population.” LASI data does not provide wealth quintile; it is MPCE quintile and they can’t be used interchangeably. So, replace the wealth quintile with the MPCE quintile, wherever used. 

Authors can avoid the limitation of using the indicator calculated based on the general population by calculating the MPCE quintile for their sample (tribal population) as well. This can be easily done using the expenditure details on food and non-food items including spending on health, education, utilities etc. given in the household dataset.

Author’s Response: Thank you for this important insight. We understand the concern raised by the editor, in this comment. The analysis was based on the individual dataset and there were several households, from where more than one individual was interviewed.

Using the household dataset for calculating MPCE might repeat (if: both husband and wife were included) or not repeat (if two or more siblings from the same household were included) those variables required for principal component analysis, which we wanted to avoid here.

3. Follow the other comments and suggestions mentioned in the attached edited copy of the manuscript.

Author’s Response: Thank you for your valuable suggestion. We have addressed suggestions made by the editor, particularly in regard to adding references, reframing the sentences, and highlighting changes done.

---

## [Editor Report · Decision Letter 3]

16 Feb 2023

Social determinants of tobacco use among tribal communities in India: evidence from the first wave of Longitudinal Aging study of India

PONE-D-22-13582R3

Dear Dr. Pati,

We’re pleased to inform you that your manuscript has been judged scientifically suitable for publication and will be formally accepted for publication once it meets all outstanding technical requirements.

Kind regards,

Chandan Kumar, Ph.D.

Academic Editor

PLOS ONE
---

## [Editor Report · Acceptance letter]

20 Feb 2023

PONE-D-22-13582R3 

Social determinants of tobacco use among tribal communities in India: evidence from the first wave of Longitudinal Ageing study in India 

Dear Dr. Pati:

I'm pleased to inform you that your manuscript has been deemed suitable for publication in PLOS ONE. Congratulations! Your manuscript is now with our production department. 

Kind regards, 

on behalf of

Dr. Chandan Kumar 

Academic Editor

PLOS ONE